# The Efficacy of Anthropometric Indicators in Predicting Non-Alcoholic Fatty Liver Disease Using FibroScan^®^ CAP Values among the Taiwanese Population

**DOI:** 10.3390/biomedicines11092518

**Published:** 2023-09-12

**Authors:** Meng-Szu Lee, Eva Belingon Felipe-Dimog, Jeng-Fu Yang, Yi-Yu Chen, Kuan-Ta Wu, Hsiang-Ju Kuo, Tzu-Chun Lin, Chao-Ling Wang, Meng-Hsuan Hsieh, Chia-Yi Lin, Batbold Batsaikhan, Chi-Kung Ho, Ming-Tsang Wu, Chia-Yen Dai

**Affiliations:** 1Department of Public Health, College of Health Sciences, Kaohsiung Medical University, Kaohsiung City 807, Taiwan; vvvykimo@gmail.com (M.-S.L.); or evafelipedimog@gmail.com (E.B.F.-D.); hochikung@yahoo.com (C.-K.H.); 2Health Management Center, Department of Occupational and Environmental Medicine, Kaohsiung Medical University Hospital, Kaohsiung Medical University, Kaohsiung City 80756, Taiwan; yang39246118@gmail.com (J.-F.Y.); id104084138@gmail.com (K.-T.W.); pipiliu@seed.net.tw (C.-L.W.); hsmonyan@gmail.com (M.-H.H.); chiai@kmu.edu.tw (C.-Y.L.); 3Nursing Department, Mountain Province State Polytechnic College, Bontoc 2616, Mountain Province, Philippines; 4Executive Master of Healthcare Administration, Department of Healthcare Administration and Medical Informatics, Kaohsiung Medical University Hospital, Kaohsiung City 80756, Taiwan; tclin1983@gmail.com; 5Hepatobiliary Division, Department of Internal Medicine, Kaohsiung Medical University Hospital, Kaohsiung Medical University, Kaohsiung City 80756, Taiwan; 6School of Medicine, College of Medicine, Kaohsiung Medical University, Kaohsiung City 80756, Taiwan; 7Department of Internal Medicine, Institute of Medical Sciences, Mongolian National University of Medical Sciences, Ulaanbaatar 14210, Mongolia; batbold_ub@yahoo.com; 8Research Center for Precision Environmental Medicine, Kaohsiung Medical University, Kaohsiung City 80756, Taiwan; 9Department of Family Medicine, Kaohsiung Medical University Hospital, Kaohsiung Medical University, Kaohsiung City 80756, Taiwan; 10Ph.D. Program in Environmental and Occupational Medicine, Kaohsiung Medical University, Kaohsiung City 80756, Taiwan; 11Drug Development and Value Creation Research Center, Kaohsiung Medical University, Kaohsiung City 87056, Taiwan

**Keywords:** non-alcoholic fatty liver disease, controlled attenuation parameters, anthropometric indicators

## Abstract

The controlled attenuation parameter (CAP) measurement obtained from FibroScan^®^ is a low-risk method of assessing fatty liver. This study investigated the association between the FibroScan^®^ CAP values and nine anthropometric indicators, including the abdominal volume index (AVI), body fat percentage (BFP), body mass index (BMI), conicity index (CI), ponderal index (PI), relative fat mass (RFM), waist circumference (WC), waist–hip ratio (WHR), and waist-to-height ratio (WHtR), and risk of non-alcoholic fatty liver disease (fatty liver). We analyzed the medical records of adult patients who had FibroScan^®^ CAP results. CAP values <238 dB/m were coded as 0 (non- fatty liver) and ≥238 dB/m as 1 (fatty liver). An individual is considered to have class 1 obesity when their body mass index (BMI) ranges from 30 kg/m^2^ to 34.9 kg/m^2^. Class 2 obesity is defined by a BMI ranging from 35 kg/m^2^ to 39.9 kg/m^2^, while class 3 obesity is designated by a BMI of 40 kg/m^2^ or higher. Out of 1763 subjects, 908 (51.5%) had fatty liver. The BMI, WHtR, and PI were found to be more strongly correlated with the CAP by the cluster dendrogram with correlation coefficients of 0.58, 0.54, and 0.54, respectively (all *p* < 0.0001). We found that 28.3% of the individuals without obesity had fatty liver, and 28.2% of the individuals with obesity did not have fatty liver. The BMI, CI, and PI were significant predictors of fatty liver. The BMI, PI, and WHtR demonstrated better predictive ability, indicated by AUC values of 0.72, 0.68, and 0.68, respectively, a finding that was echoed in our cluster group analysis that showed interconnected clustering with the CAP. Therefore, of the nine anthropometric indicators we studied, the BMI, CI, PI, and WHtR were found to be more effective in predicting the CAP score, i.e., fatty liver.

## 1. Introduction

Fatty liver disease is currently the most common liver disease in the world [1], but it is reversible, and early detection and control can prevent its progression to liver cirrhosis or liver cancer [2]. Based on a large cohort study of 130,000 adults who have yearly health examinations in Taiwan, 41% of adult men and 21% of adult women have fatty liver [3]. Obesity has been found to be a significant risk factor for this liver disease [4], and 80% of Taiwan’s fatty liver patients are obese [3], and obesity is also a risk factor for many chronic diseases [5]. The National Health and Nutrition Survey (NHNS) and the National Health Interview Survey (NHIS), both conducted in Taiwan, have reported that the prevalence of being overweight or obese among Taiwanese adults in 2017 was 29.9% and 24.9% in men and 20.4% and 16.7% in women, respectively [6].

The most widely accepted method of diagnosing fatty liver and assessing other fatty liver-associated health conditions, such as the degree of steatosis and liver fibrosis, is liver biopsy. However, there are drawbacks to liver biopsies, including their invasiveness, associated adverse outcomes, and sampling inconsistency [7,8,9,10]. Additionally, liver biopsy results may vary in response to changes in fatty liver histopathological parameters resulting from physical activity and therapeutic interventions [11]. Therefore, there is a need for non-invasive methods of diagnosing this disease. These methods include serum biomarkers and various imaging techniques, including ultrasonography, computerized tomography, magnetic resonance imaging, and elastography [12]. FibroScan^®^ is a non-invasive instrument that uses an ultrasonic probe to release a slight shock wave to the liver to directly measure the controlled attenuation parameter (CAP), a quantification of the liver fat content [13,14]. One systematic review and meta-analysis found the CAP to be sensitive in its detection of mild, moderate, and severe hepatic steatosis (87%, 85%, and 76%, respectively) [12]. That study’s results suggest that the CAP value can be used to screen for fatty liver and continually monitor liver fat content in patients. The normal reference range of the CAP value is between 100 and 238 decibels per meter (dB/m) [15]. When the value exceeds 238 dB/m, fatty liver is defined; the higher the CAP value, the more severe the degree of fatty liver [16,17]. In 2020, new terminology was introduced to describe a condition related to metabolic-associated fatty liver disease (MAFLD). The identification of MAFLD involves the detection of liver fat accumulation along with at least one of the following criteria: type 2 diabetes mellitus (T2DM), obesity, or metabolic irregularities. Notably, there has been a shift from using the term NAFLD to adopting the term MAFLD [18]. 

Fatty liver and other chronic diseases, such as obesity, diabetes, metabolic disease, and cardiovascular disease, have been found by many studies to be closely related [19,20,21]. An individual is considered to have class 1 obesity when their body mass index (BMI) ranges from 30 kg/m^2^ to 34.9 kg/m^2^. Class 2 obesity is defined by a BMI ranging from 35 kg/m^2^ to 39.9 kg/m^2^, while class 3 obesity is designated by a BMI of 40 kg/m^2^ or higher. According to the guidelines for the treatment of obesity, a BMI of 25 kg/m^2^ or higher triggers the need for further assessment in all patients. However, for individuals with genetic heritage from South Asian, Southeast Asian, and East Asian backgrounds, a BMI of 23 kg/m^2^ or more may signify the necessity for such an evaluation. This distinction is made because health risks linked to excessive weight and obesity are commonly seen at lower BMIs within these ethnic groups [22]. 

Different anthropometric measures, mostly calculated using general physical health examination data, are used to assess obesity [23,24]. These measures include the abdominal volume index (AVI), body fat percentage (BFP), body mass index (BMI), conicity index (CI), ponderal index (PI), relative fat mass (RFM), waist circumference (WC), waist–hip ratio (WHR), and waist-to-height ratio (WHtR) [25]. The relative relationship between these nine anthropometric indicators of obesity and fatty liver has not been studied. In the present study, we investigated the association between these anthropometric indices and the presence of fatty liver by studying the relationship between the values of these indices and the CAP cutoff values used to assess fatty liver. 

## 2. Materials and Methods

### 2.1. Research Sample

In this retrospective single-center study, we collected FibroScan^®^ CAP results and age, gender, height, weight, waist circumference, hip circumference, and blood pressure (SBP and DBP) data from all the patients (n = 2422) receiving FibroScan^®^ in one medical center located in a southwestern city in Taiwan from 2017 to 2020. After excluding patients with incomplete datasets, patients with chronic hepatitis B and C, and patients with hemodialysis, we were left with 1763 patients to include in our analysis. The protocol for this study was approved by the Institutional Review Board of Kaohsiung Medical University Hospital (KMUHIRB-E(I)-20190008).

### 2.2. Anthropometric Criteria Defining Obesity

Guided by previous studies [26,27,28,29], we used the following formula for calculating the results of the various indices (AVI, BFP, BMI, CI, PI, RFM, WC, WHR, WHtR) to diagnose obesity and set cutoff values. The calculation formulas are listed in Table 1.

### 2.3. Interpretation of FibroScan^®^ CAP Values for Fatty Liver Diagnosis

The diagnosis of fatty liver was based on the FibroScan^®^ CAP values. A normal liver fat level (S0) was defined if a subject had a CAP value < 238 dB/m, mild fatty liver (S1) was determined if the CAP was 238–259 dB/m, moderate fatty liver (S2) was determined if the CAP was 260–289 dB/m (S2), and severe fatty liver (S3) was determined if the CAP was ≥ 290 dB/m [30]. In the final model, the CAP values were dichotomized into having no fatty liver (CAP < 280 dB/m) and having fatty liver (CAP ≥ 238 dB/m).

### 2.4. Statistical Analysis

The normally distributed data were summarized using the mean (± standard deviation), and comparisons between groups were made using the *t*-test and ANOVA. The non-normally distributed data were summarized using their median values and compared using the Mann–Whitney and Kruskal–Wallis tests. Linear correlation analysis was employed to assess the association between the anthropometric indicators (AVI, BFP, BMI, CI, PI, RFM, WC, WHR, WHtR) and the CAP values, encompassing univariate and multivariate analyses. The overall performance of the nine anthropometric indicators in predicting NFALD (CAP cutoff values) was evaluated using the area under a receiver operating characteristic curve (AUC) reported with a 95% confidence interval (CI). The AUC values ranged between 0.5 and 1.0. The larger the AUC, the higher the diagnostic accuracy. An AUC value of 0.90–1.00 indicated excellent predictive ability, 0.80–0.90 indicated good predictive ability, 0.70–0.89 indicated fair predictive ability, 0.60–0.69 indicated weak predictive ability, and 0.50–0.59 indicated no predictive ability [31]. Multivariable logistic regression was used to build a model using the nine indicators to diagnose fatty liver. All the *p* values were two-sided and were considered significant if < 0.05. All the statistical operations were performed using Medcalc V.18 and Minitab V.19 software. 

## 3. Results

### 3.1. Characteristics of the Research Sample

We identified 2422 medical records of subjects who received FibroScan^®^ examinations at the Health Check-up Center of KMUH from 2017 to 2021. After excluding records with missing or invalid data (n = 659), we were left with the medical records of 1763 subjects to include in our analysis (Figure 1). They had a mean age of 52.2 ± 12.5 years, and 1054 (59.8%) were males (mean age of 52.5 ± 12.5 years) and 709 (40.2%) were females (mean age of 51.6 ± 12.5 years). The anthropometric data, anthropometric indicators, and CAP values of these patients were expressed as median values and grouped by the CAP grade and categorized into liver steatosis categories S0 to S3. We found significant differences in these values between all the groups (all *p* < 0.001). All the anthropometric indicators except the CI and RFM increased across the CAP. Nine hundred and eight (n = 908, 51.5%) of the patients had CAP values ≥ 238 dB/m and were, therefore, diagnosed as having fatty liver. The Mann–Whitney test showed there was a significantly higher number of subjects who had fatty liver than those who did not (*p* < 0.001) (Table 2).

#### Correlations between Anthropometric Indicators and FibroScan^®^ CAP

The Spearman correlation coefficient showed a significant positive correlation between the anthropometric indicators and the CAP value (*p* < 0.05). The correlation coefficients in decreasing order were BMI (0.58) > WHtR (0.54), PI (0.54) > WC (0.47), AVI (0.47) > WHR (0.43) > BFP (0.23) > CI (0.22) > RFM (0.12). The correlation coefficients for the BMI, WHtR, and PI were relatively high (0.58, 0.54, and 0.54, respectively) (Table 3).

A cluster dendrogram between the nine anthropometric indicators and the CAP was created using cluster analysis to calculate the correlation coefficient distance with the sum of squared deviations. We found three high-similarity pairs: AVI with WC, BMI with PI, and BFP with RFM. Three clusters were thus created according to the similarity of the data: Cluster 1 (AVI, CI, WC, and WHR), Cluster 2 (BFP and RFM), and Cluster 3 (BMI, PI, WHtR, and CAP) (Figure 2). Cluster 3 (BMI, PI, and WHtR) was highly correlated with the CAP value (Figure 2 and Table 3).

### 3.2. No Fatty Liver and Fatty Liver Groups That Fulfilled and Did Not Fulfill the Obesity Diagnostic Criteria

The difference between the no fatty liver and fatty liver groups that were not obese was significant (*p* < 0.001). A significant percentage (%) of the subjects that were not obese were found to have CAP value-diagnosed fatty liver. Almost 51% of the subjects who were not defined as obese by the CI indicator were found to have NFALD. Additionally, 45.4%, 36.6%, and 36.6% of the subjects not defined as obese by WHR, WC, and AVI indicators, respectively, were also found to have NFALD. The median CAP values in the subjects with fatty liver who were not obese ranged from CAP 262 to 279 dB/m, indicating moderate fatty liver. However, in general, more of those who were not obese and did not have fatty liver than those who were also not obese and did have the disease had a much higher rate of no fatty liver compared to their rates of fatty liver (Table 4).

The obese subjects with fatty liver had significantly higher anthropometric indicator values (all) than the obese subjects with no fatty liver. About 72% of the subjects defined as obese based on their BMI value were found to have NFALD, while 28.2% of them had no fatty liver. Similarly, 71.7%, 71.0%, and 69.1% of the subjects defined as obese by the WHR, WHtR, and AVI, respectively, had a higher frequency of fatty liver. The obese subjects with fatty liver had median CAPs ranging from CAP 280 to 289 dB/m, indicating moderate fatty liver. In addition, the Mann–Whitney test showed a significant difference (*p* < 0.001) between individuals who did and did not meet the obesity diagnostic criteria for obesity. For the obese samples, the Mann–Whitney test showed that the number of people with fatty liver was significantly higher than those without (Table 4).

### 3.3. Anthropometric Indicators to Predict Fatty Liver

In our logistic regression analysis, the BMI, CI, and PI were found to be significant predictors of fatty liver. With each unit increase in the BMI score, there was about a three-fold increase in the likelihood that a subject would have fatty liver (AOR 2.65, 95% CI 1.93–3.64). Similarly, with every unit increase in the CI score, there was a 0.66 increase in the likelihood of fatty liver (AOR 0.66, 95% CI 0.480.90). Finally, each unit increase in the PI score came with a 1.60 (AOR 1.69, 95% CI 1.22–2.34) increase in the risk of having fatty liver (1.69) (Table 5).

The efficacy of the diagnostic quasi-values of each indicator was evaluated by a ranking of AUC values. AUC analysis revealed that the BMI (AUC 0.72, 95% CI 0.70–0.74) had the highest predictive ability of the nine indicators, followed by the PI (AUC 0.68, 95% CI 0.66–0.71) and WHtR (AUC 0.68, 95% CI 0.66–0.70) (Figure 3). This result was in accordance with the same group in the dendrogram in our cluster group analysis, which revealed the BMI, PI, and WHtR to be interconnectedly clustered with the CAP (Figure 3).

## 4. Discussion

This study found all nine anthropometric indicators (AVI, BFP, BMI, CI, PI, RFM, WC, WHR, and WHtR) to be significantly and positively correlated with the FibroScan^®^ CAP values. The BMI, PI, and WhtR had the most significant correlation coefficients. Taking a CAP ≥ 238 dB/m as the standard for diagnosing fatty liver, a comparison between no NFALD and NFALD subjects that met or did not meet the obesity diagnostic criteria was assessed. Using a CAP ≥ 238 dB/m as a cutoff value, we found that a little more than 50% of the subjects had fatty liver. This rate was higher than those reported in previous studies, which found the prevalence of fatty liver among Taiwanese adults to be 11.5% in 2016 and 44.5% in 2020 [32,33]. It is also higher than the estimated worldwide prevalence of 25.2% [4], a rate suggesting the need for public health intervention programs. Although our subjects had a median CAP value, which indicated mild to moderate fatty liver, there is a need to improve our ability to detect this disease earlier and develop strategies to prevent it in Taiwan. 

This study also found a significant correlation between the nine indicators and the CAP values. The top three indicators positively correlated with the CAP were the BMI (r = 0.58), WHtR (r = 0.54), and PI (r = 0.54). This result was consistent with our cluster analysis, which showed a significant and highly correlated relationship between the CAP value and the BMI, PI, and WHtR (*p* < 0.05). The BMI, PI, and WHtR values increased concurrently with increases in the CAP values. One recent study using the CAP to assess steatosis in 104 participants with a mean BMI of 30.4 kg/m^2^ found them to have scores indicating S2 (moderate) or S3 (severe steatosis), suggesting a positive correlation between the BMI and fatty liver [34]. Their findings and ours suggest that increases in the BMI and other anthropometric indicators can be used to predict an increase in the CAP. The BMI and WC, often used as indicators of obesity, have been associated with fatty liver [35]. The WHtR, a new obesity-related anthropometric indicator, has previously been found to predict hypertension, stroke, and dyslipidemia [36], but until this study, it had not been related to fatty liver. 

This study found that a significant number of non-obese subjects had fatty liver. Other studies have also found that people with liver disease are not necessarily obese [4,37]. Two studies have proposed that being overweight or obese are risk factors and not causative factors of fatty liver [38,39]. On the other hand, a significant number of obese subjects did not have fatty liver, indicating that some obese subjects do not necessarily have fatty liver. One study found the prevalence of fatty liver in an obese population to be about 90% [40], leaving 10 percent of obese individuals to be fatty liver free. This disease-free state may be explained by the fact that some obese individuals may have good blood circulation because of their involvement in physical activities, which can contribute to a decreased risk of fatty liver. In fact, another study found decreased hepatic steatosis and liver inflammation in individuals with fatty liver who participated in about 2.5 h of mildly moderate exercise [41]. Because of this, there might be a need to adjust the obesity diagnostic criteria for each indicator to avoid misdiagnosis. The result also suggests that several of the obesity diagnostic criteria of the anthropometric indicators may either be poor predictors of fatty liver or that new cutoff values for each indicator are required to predict fatty liver. 

We further analyzed the relationship between the anthropometric indicators and fatty liver using multivariable logistic regression. The BMI, CI, and PI were found to be significant predictors of NFALD, a finding similar to those of previous studies that also found the BMI to be an important indicator able to predict fatty liver [42,43]. The CI [44] and PI [45] are also anthropometric indicators of obesity. We found them to be predictive of fatty liver. This demonstrates that obesity is correlated with NFALD, as previous studies have found [39,46]. 

This paper has some limitations. One limitation is that it has a limited number of research samples obtained from one medical center. Another limitation is that we did not take into consideration the influence of gender and age. Furthermore, we believed that the study results might be only applied to Taiwanese people. The BMI is influenced by muscle mass, which was not available in the present study. In addition, the body impedance, data on alcohol consumption, or statin therapy were not provided for the individuals. Further multi-center studies or nationwide studies could be conducted to confirm our findings before our results could be applied to the clinical diagnosis of fatty liver and the assessment of the severity of the disease.

## 5. Conclusions

In conclusion, this study found all nine anthropometric indicators (AVI, BFP, BMI, CI, PI, RFM, WC, WHR, and WHtR) to be significantly and positively correlated with the FibroScan^®^ CAP values. The BMI, PI, and WhtR were found to have the most significant correlation coefficients. The BMI, CI, and PI consistently predicted fatty liver. Not all of our subjects categorized as obese had fatty liver and not all of the fatty liver subjects were obese, suggesting a need for further adjustment of the obesity diagnostic criteria of the anthropometric indicators and new CAP cutoff values for each indicator to better predict fatty liver. 

## Figures and Tables

**Figure 1 biomedicines-11-02518-f001:**
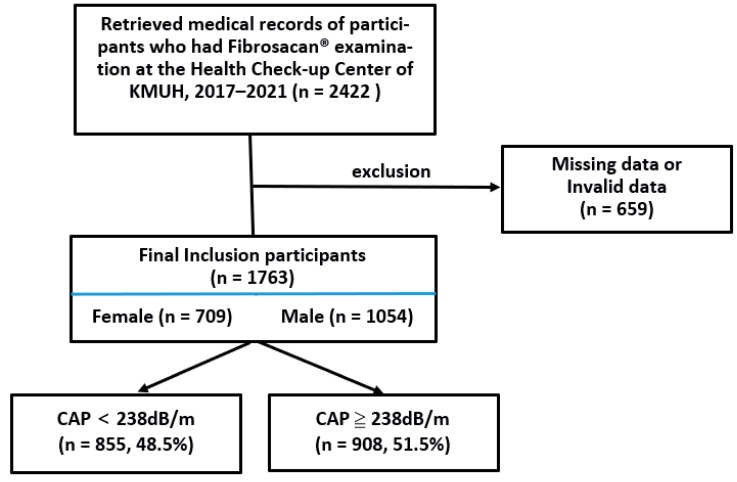
Flow chart of the patients in the present study.

**Figure 2 biomedicines-11-02518-f002:**
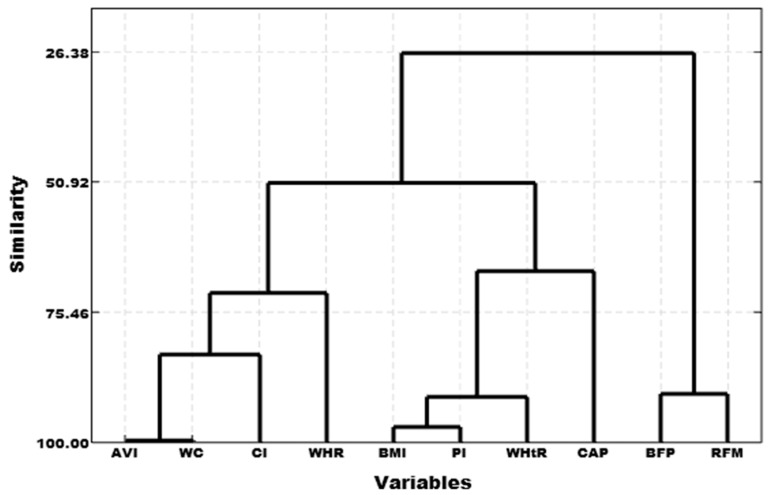
Cluster dendrogram of anthropometric indicators and CAP values.

**Figure 3 biomedicines-11-02518-f003:**
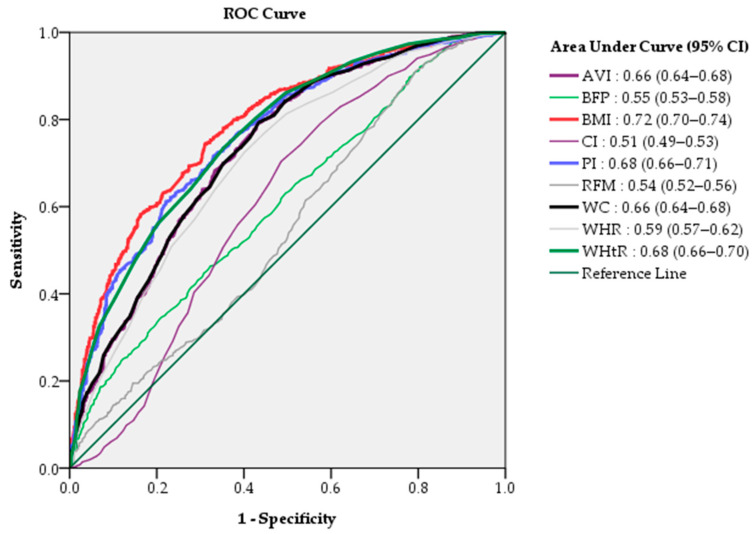
The relative ability of the nine anthropometric measurement values to predict non-alcoholic fatty liver assessed by AUC.

**Table 1 biomedicines-11-02518-t001:** Formula of anthropometric indices and criteria for diagnosing obesity.

Anthropometric Indices	Formula	Diagnostic Criteria for Obesity
Abdominal volume index (AVI)	AVI = [2WC^2^ (cm) + 0.7(WC − Height)^2^ (cm)]/1000;	≥17.49
Body fat percentage (BFP)	Male BFP = (1.2 × BMI) + (0.23 × Age) − 16.2 Female BFP = (1.2 × BMI) + (0.23 × Age) − 5.4	Male > 25% Female > 30%
Body mass index (BMI)	Weight (kg)/(Height(m))^2^	≥30 kg/m^2^
Conicity index (CI)	$CI=\frac{{WC}_{\left(m\right)}}{0.109\times \sqrt{\frac{{Weight}_{\left(kg\right)}}{{Height}_{\left(m\right)}}}}$	Male > 1.25 Female > 0.83
Ponderal Index (PI)	Weight (kg)/(Height (m))^3^	>17 kg/m^3^
Relative fat mass (RFM)	Male RFM = 64 − 20 × (Height(cm)/WC(cm)) Female RFM = 76 − 20 × (Height(cm)/WC(cm))	Male > 25 Female > 32
Waist circumference (WC)	-	Male ≥ 90 cm Female ≥ 85 cm
Waist-to-hip ratio (WHR)	WC(cm)/Hip(cm)	Male ≥ 0.90 Female ≥ 0.80
waist-to-height ratio (WHtR)	WC(cm)/Height(cm)	>0.5

**Table 2 biomedicines-11-02518-t002:** Subjects’ demographic, physical examination, and anthropometric indicator data categorized into degrees of liver steatosis (S0, S1, S2, and S3) and categorized into those with steatosis (CAP ≥ 234 dB/m) and without steatosis (< 238 dB/m) defined by CAP cutoff values.

Variables	All Subjects (N = 1763)	Liver Steatosis Categories ^a^					Subjects with and without Liver Steatosis ^b^		
		S0 [<238 dB/m]	S1 [238–259 dB/m]	S2 [260–289 nB/m]	S3 [≥290 dB/m]	*p* Value	CAP [<238 dB/m]	CAP [≥238 dB/m]	*p* Value
		n = 872 (49.5%)	n = 242 (13.7%)	n = 277 (15.7%)	n= 372 (21.1%)		n = 855 (48.5%)	n = 908 (51.5%)	
Gender, n (%)									
Female (mean age: 51.6 ± 12.5 years)	709 (40.2)	434 (61.2%)	79 (11.1%)	85 (12.0%)	111 (15.7%)	<0.001	429 (60.5%)	280 (39.5%)	<0.001
Male (mean age: 52.5 ± 12.5 years)	1054 (59.8)	438 (41.6%)	163 (15.5%)	192 (18.2%)	261 (24.8%)		426 (40.4%)	628 (59.6%)	
Physical examination data, median (standard deviation)									
Age (year)	53 (43–62)	51 (40–62)	54 (46–64)	56 (47–62)	53 (45–62)	<0.001	51 (40–62)	54 (45–63)	<0.001
Height (centimeter)	166 (160–172)	165 (159–171)	168 (161–173)	167 (161–172)	167 (161–173)	<0.001	165 (159–171)	167 (161–173)	<0.001
Weight (kilogram)	67 (58–76)	61 (54–69)	69 (60–76)	72 (64–80)	76 (68–86)	<0.001	61 (54–69)	73 (65–81)	<0.001
Hip (centimeter)	94 (88–98)	91 (86–95)	95 (89–98)	97 (92–101)	98 (94–104)	<0.001	91 (86–95)	96 (91–101)	<0.001
DBP (mm/hg)	75 (69–83)	73 (67–81)	75 (68–82)	78 (71–85)	79 (72–87)	<0.001	73 (67–81)	77 (71–85)	<0.001
SBP (mm/hg)	124 (114–136)	121 (110–133)	123 (112–138)	126 (117–137)	129 (119–139)	<0.001	121 (110–133)	127 (117–138)	<0.001
Anthropometric indicators, median (standard deviation)									
AVI	14.6 (12.3–17.2)	13.2 (10.9–15.6)	14.9 (13.2–17.0)	15.9 (14.0–18.5)	16.6 (14.6–19.6)	<0.001	13.1 (10.9–15.5)	15.9 (14.0–18.4)	<0.001
BFP	29.0 (25.3–33.8)	28.0 (24.2–32.4)	28.5 (25.0–33.3)	30.3 (26.0–34.8)	31.0 (27.0–37.7)	<0.001	28.0 (24.2–32.4)	30.0 (26.1–35.5)	<0.001
BMI	24.3 (21.9–26.7)	22.5 (20.7–24.6)	24.6 (22.7–26.4)	25.7 (23.9–28.0)	27.1 (24.8–29.5)	<0.001	22.5 (20.7–24.6)	25.8 (24.0–28.1)	<0.001
CI	1.22 (1.16–1.29)	1.19 (1.13–1.28)	1.23 (1.17–1.30)	1.23 (1.19–1.30)	1.24 (1.19–1.30)	<0.001	1.19 (1.13–1.28)	1.24 (1.18–1.30)	<0.001
PI	14.6 (13.3–16.1)	13.7 (12.6–14.9)	14.6 (13.7–15.8)	15.5 (14.4–16.9)	16.3 (15.0–17.9)	<0.001	13.7 (12.6–14.9)	15.5 (14.3–17.1)	<0.001
RFM	28.2 (24.4–34.1)	28.0 (23.6–33.6)	27.1 (24.1–32.7)	28.5 (25.2–33.8)	29.2 (26.0–36.2)	<0.001	28.0 (23.6–33.6)	28.5 (25.0–34.7)	<0.001
WC	85.0 (78.0–92.4)	81 (73–88)	86 (81–92)	89 (83–96)	91 (85–99)	<0.001	80.0 (73.0–88.0)	89.0 (83.3–96.0)	<0.001
WHR	0.87 (0.82–0.92)	0.85 (0.79–0.89)	0.88 (0.85–0.92)	0.89 (0.85–0.94)	0.91 (0.87–0.95)	<0.001	0.84 (0.79–0.89)	0.90 (0.86–0.94)	<0.001
WHtR	0.50 (0.46–0.54)	0.48 (0.44–0.51)	0.51 (0.47–0.53)	0.52 (0.49–0.56)	0.54 (0.50–0.57)	<0.001	0.47 (0.44–0.51)	0.52 (0.49–0.56)	<0.001

Note: ^a^ The Kruskal–Wallis test was used to compare groups of non-normally distributed data; ^b^ Mann–Whitney test was used for normally distributed data; dB/m: decibels per meter; CAP: controlled attenuation parameter; AVI: abdominal volume index; BFP: body fat percentage; BMI: body mass index; CI: conicity index; PI: ponderal index; RFM: relative fat mass; WC: waist circumference (WC): WHR: waist–hip ratio; WHtR: waist-to-height ratio.

**Table 3 biomedicines-11-02518-t003:** Spearman correlation coefficients between anthropometric indicators and CAP values.

	CAP	AVI	BFP	BMI	CI	PI	RFM	WC	WHR	WHtR
**CAP**	1.00	<0.0001	<0.0001	<0.0001	<0.0001	<0.0001	<0.0001	<0.0001	<0.0001	<0.0001
**AVI**	0.47	1.00	<0.0001	<0.0001	<0.0001	<0.0001	<0.0001	<0.0001	<0.0001	<0.0001
**BFP**	0.23	0.20	1.00	<0.0001	<0.0001	<0.0001	<0.0001	<0.0001	<0.0001	<0.0001
**BMI**	0.58	0.73	0.40	1.00	<0.0001	<0.0001	<0.0001	<0.0001	<0.0001	<0.0001
**CI**	0.22	0.80	0.11	0.27	1.00	<0.0001	<0.0001	<0.0001	<0.0001	<0.0001
**PI**	0.54	0.60	0.60	0.93	0.22	1.00	<0.0001	<0.0001	<0.0001	<0.0001
**RFM**	0.12	0.28	0.82	0.21	0.37	0.42	1.00	<0.0001	0.001	<0.0001
**WC**	0.47	1.00	0.19	0.73	0.80	0.60	0.27	1.00	<0.0001	<0.0001
**WHR**	0.43	0.63	−0.01	0.55	0.54	0.44	−0.08	0.65	1.00	<0.0001
**WHTR**	0.54	0.76	0.46	0.84	0.54	0.83	0.37	0.77	0.74	1.00

Note: AVI: abdominal volume index; BFP: body fat percentage; BMI: body mass index; CI: conicity index; PI: ponderal index; RFM: relative fat mass; WC: waist circumference (WC): WHR: waist–hip ratio; WHtR: waist-to-height ratio.

**Table 4 biomedicines-11-02518-t004:** The anthropometric indicator values of non-obese and obese subjects with no fatty liver (CAP < 238 dB/m) and those with fatty liver (CAP ≥ 238 dB/m).

Variables	Did Not Meet Obesity Diagnostic Criteria					Met Obesity Diagnostic Criteria				
	No Fatty Liver		Fatty Liver			No Fatty Liver		Fatty Liver		
	n (%)	Median (s.d.)	n (%)	Median (s.d.)	*p* Value	n (%)	Median (s.d.)	n (%)	Median (s.d.)	*p* Value
AVI	605 (63.4%)	203 (185–219)	349 (36.6%)	267 (251–301)	<0.001	250 (30.9%)	212 (195–228)	559 (69.1%)	286 (261–319)	<0.001
BFP	142 (71.7%)	203 (188–218)	56 (28.3%)	266 (246–288)	<0.001	713 (45.6%)	207 (188–222)	852 (54.4%)	281 (256–314)	<0.001
BMI	589 (71.7%)	201 (184–218)	232 (28.3%)	262 (247–285)	<0.001	266 (28.2%)	215 (200–227)	676 (71.8%)	286 (261–319)	<0.001
CI	681 (49.2%)	206 (188–221)	704 (50.8%)	279 (255–314)	<0.001	174 (46.0%)	203 (188–223)	204 (54.0%)	283 (258–314)	<0.001
PI	485 (72.8%)	199 (182–217)	181 (27.2%)	264 (247–285)	<0.001	370 (33.7%)	213 (197–226)	727 (66.3%)	284 (259–316)	<0.001
RFM	92 (79.3%)	201 (180–215)	24 (20.7%)	257 (248–280)	<0.001	763 (46.3%)	206 (189–222)	884 (53.7%)	280 (256–314)	<0.001
WC	606 (63.4%)	204 (186–220)	350 (36.6%)	276 (254–305)	<0.001	249 (30.9%)	212 (196–228)	558 (69.1%)	286 (261–319)	<0.001
WHR	740 (54.6%)	204 (186–220)	616 (45.4%)	276 (254–305)	<0.001	115 (28.3%)	219 (201–229)	292 (71.7%)	289 (262–331)	<0.001
WHtR	618 (65.4%)	202 (185–218)	327 (34.6%)	267 (250–296)	<0.001	237 (29.0%)	215 (198–228)	581 (71.0%)	286 (261–320)	<0.001

Note: fatty liver means CAP ≥ 238 dB/m; No fatty liver is CAP < 238 dB/m; s.d.: standard deviation; dB/m: decibels per meter; AVI: abdominal volume index; BFP: body fat percentage; BMI: body mass index; CI: conicity index; PI: ponderal index; RFM: relative fat mass; WC: waist circumference (WC): WHR: waist–hip ratio; WHtR: waist-to-height ratio.

**Table 5 biomedicines-11-02518-t005:** Logistic regression analysis of anthropometric indicators related to the risk of fatty liver.

Anthropometric Indicators	COR (95% CI)	*p* Value	AOR (95% CI)	*p* Value
AVI	1.91 (1.25–1.33)	<0.001	1.63 (0.07–39.97)	0.764
BFP	1.07 (1.05–1.09)	<0.001	0.87 (0.57–1.33)	0.533
BMI	1.42 (1.37–1.48)	<0.001	2.67 (1.94–3.66)	<0.001
CI	8.01 (7.12–25.69)	<0.001	0.66 (0.48–0.90)	0.008
PI	1.64 (1.55–1.74)	<0.001	1.69 (1.22–2.34)	0.001
RFM	1.03 (1.02–1.04)	<0.001	1.54 (0.89–2.69)	0.125
WC	1.09 (1.08–1.10)	<0.001	1.22 (0.05–29.73)	0.905
WHR	7.56 (5.73–18.36)	<0.001	1.26 (0.93–1.71)	0.134
WHtR	4.01 (3.96–11.15)	<0.001	1.28 (0.94–1.73)	0.113

Note: COR: Crude odds ratio; AOR: Adjusted odds ratio; 95% CI: 95% confidence interval; AVI: abdominal volume index; BFP: body fat percentage; BMI: body mass index; CI: conicity index; PI: ponderal index; RFM: relative fat mass; WC: waist circumference (WC): WHR: waist–hip ratio; WHtR: waist-to-height ratio.

## Data Availability

All the data used in this study are available from the corresponding author.

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
