# Peer review of "The Efficacy of Anthropometric Indicators in Predicting Non-Alcoholic Fatty Liver Disease Using FibroScan^®^ CAP Values among the Taiwanese Population"

_biomedicines, 2023, doi:10.3390/biomedicines11092518_

Round 1
Reviewer 1 Report
The paper of Lee et al. about the anthropometric indicators in prediction of non alcoholic fatty liver disease using fibroscan CAP values is well written and understandable.
The In the house does ethnicity from type 1 on yes no Russ rightauthors analyzed different anthropometric indicators in relation to measured CAP values Which were obtained in an urban healthcare facility. The most predicting indicator of non alcoholic liver disease was not unsurprisingly BMI.
There are some issues with this paper.
First: the patients coming to the healthcare facility to get a CAP measurement did this probably for a reason. Unfortunately, there are no data provided about the indications for the CAP measurements. In other words: were there concomitant liver diseases like hepatitis C that can cause an elevated CAP value? That can influence the predictive value of BMI for instance.
Second: the best area under the curve was 0.73 this is not quite impressive it doesn't give additional information other than what we know that elevated BMI is a risk factor for non alcoholic fatty liver disease.
Third: since the study was done in Taiwan the ethnicity of the analyzed population was probably pure Asian. It would be interesting to compare the results with data from other ethnicities.
Fourth: The BMI is influenced not only by body fat also by muscle mass. There are no data provided about the muscle mass of the individuals this elevated BMI and no non alcoholic fatty liver disease or non alcoholic fatty liver disease without elevated BMI.
Fifth: are there data out about body impedance measurements with this population or at least the data should be compared to eventually purplished data on the correlation of CAP and about the impedance measurements.
Taken together unfortunately the data are to crude to add a meaningful tool in the prediction of non alcoholic fatty liver disease.
Author Response
First: the patients coming to the healthcare facility to get a CAP measurement did this probably for a reason. Unfortunately, there are no data provided about the indications for the CAP measurements. In other words: were there concomitant liver diseases like hepatitis C that can cause an elevated CAP value? That can influence the predictive value of BMI for instance.
Thanks for the comments. The study was conducted at the Health Check-up Center of KMUH. We have excluded all patients with chronic hepatitis B and C in the present study. (page 3, line 114).
Second: the best area under the curve was 0.73 this is not quite impressive it doesn't give additional information other than what we know that elevated BMI is a risk factor for non alcoholic fatty liver disease.
Thank you very much for the comment. In this large cohort study, we collected relatively healthy individuals with the data of the Fibroscan. The result of the AUROC of BMI reflected the importance with the reports of Fibroscan.
Third: since the study was done in Taiwan the ethnicity of the analyzed population was probably pure Asian. It would be interesting to compare the results with data from other ethnicities.
Thank you very much for the comment. We have added the limitation that the results only applied in Taiwanese people. (page 9, line 289-290)
Fourth: The BMI is influenced not only by body fat also by muscle mass. There are no data provided about the muscle mass of the individuals this elevated BMI and no non alcoholic fatty liver disease or non alcoholic fatty liver disease without elevated BMI.
Thank you very much for the comment. We have added the limitation that we do not have the data of the muscle mass. (page 9, line 290-291)
Fifth: are there data out about body impedance measurements with this population or at least the data should be compared to eventually purplished data on the correlation of CAP and about the impedance measurements.
Thank you very much for the important comment. Since this is a retrospective study, we do not have the data of body impedance measurements and we have added this point as a limitation. (page 9, line 291)
Reviewer 2 Report
I have read and analyzed the manuscript from Lee and coauthors. I think that the present study is interesting and potentially has significant clinical impact. However, the present manuscript does not contain any biological mechanisms, innovative translational research or something like that. Put it otherwise, the manuscript does not adjust to the aims and scope of Biomedicine. I want to suggest to the Editor transfer the manuscript in other clinical MDPI journals such as Journal of Clinical Medicine with saving of our review and continue the discussion in the framework of other journal.
Critical points:
1.Fibroscan is a method based on ultrasonic signals. However, fat is the tissue which has problems with echogenicity and fat is the limitation in ultrasonography of the abdominal organs. Is this fact a limitation of Fibroscan liver fat evaluation?
2.I do not see any data about patients' informed consent. Did authors obtain agreement from patients for the use of personal clinical data in the study?
3.Did authors analyze the history of alcohol consumption or statin therapy? Both of these factors can critically influence liver health and disease.
4.The most critical question. Authors declared domestic specific criteria of obesity. First of all, the Taiwanese population as Asians has specific metabolic features. Secondly, how are selected criteria adjusted with ADA or AHA obesity criteria? Moreover, in my opinion, authors just create their own artificial criteria of obesity which allows correlating Fibroscan parameters with BMI and other selected parameters.
5.Which post hoc test was used for Kruskel Wallis test?
6.I think that the authors determine the results of correlation analysis incorrectly. In Table 2 correlation coefficient between WC and WC is 1, it is normal. But why correlation between WC and AVI also 1?
7.Correlation analysis. Where are p values for all coefficients? I’m not confident that for correlation coefficient, for instance, 0.12 p value less than 0.05.
8.I suggest to authors add body weight to anthropometric measurements.
9.For nine selected parameters authors should add a graph with correlation trend line with all used points. At least, this addition can be performed in Supplement.
10.The study has significantly more limitations than 1. This subsection should be updated.
Author Response
1. Fibroscan is a method based on ultrasonic signals. However, fat is the tissue which has problems with echogenicity and fat is the limitation in ultrasonography of the abdominal organs. Is this fact a limitation of Fibroscan liver fat evaluation?
Thank you very much for the comment. The fibroscan has been considered as an important non-invasive measurement for grading of steatosis. We have also published the paper in the previous issue of the Biomedicine (Ref. 13)
2.I do not see any data about patients' informed consent. Did authors obtain agreement from patients for the use of personal clinical data in the study?
Thank you very much for the comment. Since this is a retrospective study, we have the approval of the Institutional Review Board of Kaohsiung Medical University Hospital (KMUHIRB-E(I)-20190008) as stated in the text. (page 3 line 115-117)
3.Did authors analyze the history of alcohol consumption or statin therapy? Both of these factors can critically influence liver health and disease.
Thank you very much for the comment. It is another limitation in our study that no data on alcohol consumption or statin therapy and we have added this point as a limitation. (page 9, line 290-291).
4.The most critical question. Authors declared domestic specific criteria of obesity. First of all, the Taiwanese population as Asians has specific metabolic features. Secondly, how are selected criteria adjusted with ADA or AHA obesity criteria? Moreover, in my opinion, authors just create their own artificial criteria of obesity which allows correlating Fibroscan parameters with BMI and other selected parameters.
Thank you very much for the comment. In the present study, we have one of the largest cohorts with Fibroscan data. With this advantage, we may have the chance to re-identify the criteria of obesity from the point of view of fatty liver. We have revised the abstract (page 1 line 41-43).
5.Which post hoc test was used for Kruskel Wallis test?
Thank you very much for the comment. In this article, the Kruskal–Wallis test was used because the sample data we analyzed was tested to be a non-normal distribution. When doing the Kruskal–Wallis test, we performed Dunn's post hoc comparisons when there were significant differences in the data groups. We use it to get the results of Table 2.
6.I think that the authors determine the results of correlation analysis incorrectly. In Table 2 correlation coefficient between WC and WC is 1, it is normal. But why correlation between WC and AVI also 1?
Thank you very much for the comment. The AVI calculation formula is mainly composed of WC which might be the explanation of the correlation coefficient equal to 1. Correlation analysis (Table 3 for details) is mainly to look at the correlation between each anthropometric index and Fibroscan CAP, which is helpful to judge whether it is beneficial to incorporate into the prediction model.
7.Correlation analysis. Where are p values for all coefficients? I’m not confident that for correlation coefficient, for instance, 0.12 p value less than 0.05.
Thank you very much for the comment. We have revised Table 3 showing all the P values clearly.
8.I suggest to authors add body weight to anthropometric measurements.
Thank you very much for the comment. The body weight was analyzed in the clinical variable as the upper part of Table 2. The anthropometric measurements use the calculation of the variables. Hence, we use BMI in this part. Body weight was analyzed among the clinical variables in the upper part of Table 2. Body weight is also used as a calculation parameter in the calculation variables of anthropometric indicators, such as BMI, PC, CI, etc.
9.For nine selected parameters authors should add a graph with correlation trend line with all used points. At least, this addition can be performed in Supplement.
Thank you very much for the comment. From the Spearman correlation coefficient between anthropometric indicators and CAP values in Table 3 and the cluster dendrogram of anthropometric indicators and CAP values in Figure 1, we can clearly see the correlation trends between the nine anthropometric indicators and Fibroscan CAP.
10.The study has significantly more limitations than 1. This subsection should be updated.
Thanks for the important comment. We have revised it with some additional limitations. Thank you very much again
Reviewer 3 Report
The study by Meng-Szu Lee et al. is a retrospective single-center study evaluating 1763 subjects with a controlled attenuation parameter (CAP) measurement by FibroScan® from 2017 to 2021.
However, major changes are necessary to avoid wrong conclusions and to be accepted into biomedicines
Major comments:
1. General comment. Diagnosis of NAFLD is histologic. Therefore, authors should use the definition of MAFLD (DOI: 10.1016/j.jhep.2020.03.039; and DOI: 10.1007/s12020-022-03157-x) to avoid wrong conclusions and rewrite the study to identify patients with MAFLD, their CAP values, and the anthropometric indicators.
2. Abstract. Please, include the definition of obese and rates. Please, include the rate of misclassified patients. Please include the multivariate results with the OR, 95%CI, and “p” value for each independent variable.
3. Introduction. Please, include the definition of obesity and references. Please, use Transient elastography (TE) or FibroScan® better than fibroscan. Please, include the definition of MAFLD and references
4. Material and methods. Please, include information about the study’s nature (it is a retrospective single-center study). Please, include the references of each formula. Please include references to define obesity. In the statistical analysis, please include the univariate and multivariate analysis to identify independent variables associated with elevated CAP
5. Results. Please, include a more detailed period with months. authors should include other important variables for diagnosing MAFLD (DOI: 10.1016/j.jhep.2020.03.039). Authors should define the exclusion criteria. A Flow chart figure is necessary for a better understanding.
6. Tables and discussion. Please, change the results and the discussion according to the general recommendation.
Author Response
1. General comment. Diagnosis of NAFLD is histologic. Therefore, authors should use the definition of MAFLD (DOI: 10.1016/j.jhep.2020.03.039; and DOI: 10.1007/s12020-022-03157-x) to avoid wrong conclusions and rewrite the study to identify patients with MAFLD, their CAP values, and the anthropometric indicators.
Thank you very much for the comment. We completely agree with the opinion of the Reviewer that MAFLD may be the main description. In the retrospective study, we do not have the data on parameters of the MAFLD such as insulin and CRP levels. We would like to use” fatty liver” only for the present study which might make sense and avoid the controversy. We have added the description of MAFLD (page 2 line 83-87)
2. Please, include the definition of obese and rates. Please, include the rate of misclassified patients. Please include the multivariate results with the OR, 95%CI, and “p” value for each independent variable.
Thank you very much for the comment. We have revised the abstract (page 1 line 41-43) according to the comment.
3. Introduction. Please, include the definition of obesity and references. Please, use Transient elastography (TE) or FibroScan® better than fibroscan. Please, include the definition of MAFLD and references.
Thank you very much for the comment. We have revised the Introduction (page 2 line 90-98) according to the comment.
4. Material and methods. Please, include information about the study’s nature (it is a retrospective single-center study). Please, include the references of each formula. Please include references to define obesity. In the statistical analysis, please include the univariate and multivariate analysis to identify independent variables associated with elevated CAP.
Thank you very much for the comment. We have revised the Material and methods (page 3 line 110, line 119, line 134-136 )
5. Results. Please, include a more detailed period with months. authors should include other important variables for diagnosing MAFLD (DOI: 10.1016/j.jhep.2020.03.039). Authors should define the exclusion criteria. A Flow chart figure is necessary for a better understanding.
Thank you very much for the comment. We completely agree with the opinion of the Reviewer that MAFLD may be the main description. In the retrospective study, we do not have the data on parameters of the MAFLD such as insulin and CRP levels. We would like to use” fatty liver” only for the present study which might make sense and avoid the controversy.
6. Tables and discussion. Please, change the results and the discussion according to the general recommendation.
Thank you very much for the comment. We have revised the manuscript.
Round 2
Reviewer 1 Report
With the limitations clarified the paper has been improved.
Author Response
With the limitations clarified the paper has been improved.
Ans: Thank you very much for the help.
Reviewer 2 Report
Many thanks to the authors for the response. However, responses on some questions are not completed.
1. Ok, but how it can explain the problem with fat echogenicity? The fact of author's previous study publication is not an evidence.
2. Ok.
3. Ok.
4. I'm not agree. Do ADA or AHA (or any other large professional community) change obesity criteria in NAFLD context? Or what is the hepatologists opinion about specific NAFLD/obesity joint criteria?
5. Ok.
6. Ok.
7. Ok.
8. Ok.
9. Ok.
10. Ok.
Author Response
- Ok, but how it can explain the problem with fat echogenicity? The fact of author's previous study publication is not an evidence.
Ans: Thanks for the comment. The development of transient elastography of the controlled attenuation parameter (CAP) (Fibroscan) has allowed simultaneous and reasonably accurate assessment of hepatic steatosis and fibrosis by hepatologists (Clin Mol Hepatol. 2020 Apr;26(2):128-141. doi: 10.3350/cmh.2019.0001n). The European Association of Study of the Liver (EASL) has also recommended the Fibroscan CAP as another imaging technique to diagnose steatosis. (J Hepatol. 2016 Jun;64(6):1388-402. doi: 10.1016/j.jhep.2015.11.004.)
- Ok. Ans: Thank you very much.
- Ok. Ans: Thank you very much.
- I'm not agree. Do ADA or AHA (or any other large professional community) change obesity criteria in NAFLD context? Or what is the hepatologist's opinion about specific NAFLD/obesity joint criteria?
Ans: Thanks for the comment. We completely agree with the reviewer’s opinion and are sorry for the misunderstanding. In the abstract, we have added the criteria of obesity which is exactly the criteria of ADA and AHA. We did not change the obesity criteria at all. We believe also that the specific NAFLD/obesity joint criteria need to be studied further and created with more consent.
- Ok. Ans: Thank you very much.
- Ok. Ans: Thank you very much.
- Ok. Ans: Thank you very much.
- Ok. Ans: Thank you very much.
- Ok. Ans: Thank you very much.
- Ok. Ans: Thank you very much.
Thank you very much for reviewing our manuscript again, Dear Reviewer.
Reviewer 3 Report
The study by Meng-Szu Lee et al. is a retrospective single-center study evaluating 1763 subjects with a controlled attenuation parameter (CAP) measurement by FibroScan® from 2017 to 2021. The authors have provided a revised version. However, some minor changes are necessary to avoid wrong conclusions and to be accepted into biomedicines
1. Title. Due to the anthropometric differences related to ethnic groups, authors should include “in Taiwanese people” in the title
2. Abstract. Please, include the values and rates in the results and avoid terms as “some” or “while some” or “were more” without the results. Please include the OR, 95%CI, and “p” value for each independent variable.
3. Please, use FibroScan® better than fibroscan (title, lines 34, 36, and 40)
4. Results. Please, include a more detailed period of time with the months. Authors should include a figure with the flow chart of included/excluded patients for a better understanding.
Author Response
- Due to the anthropometric differences related to ethnic groups, authors should include “in Taiwanese people” in the title
Ans: Thank you very much for the comment. We have revised the title to “The efficacy of anthropometric indicators in predicting non-alcoholic fatty liver disease using Fibroscan® CAP values among the Taiwanese population”.
- Please, include the values and rates in the results and avoid terms as “some” or “while some” or “were more” without the results. Please include the OR, 95%CI, and “p” value for each independent variable.
Ans: Thank you very much for the comment. We have revised the abstract as the reviewer’s comment.
- Please, use FibroScan® better than fibroscan (title, lines 34, 36, and 40)
Ans: Thank you very much for the comment. We are sorry for the mistake of not revising all the fibroscan as FibroScan®, and we have revised all currently.
- Results. Please, include a more detailed period of time with the months. Authors should include a figure with the flow chart of included/excluded patients for a better understanding.
Ans: Thank you very much for the comment. We have added Figure 1 with the flow chart as the reviewer’s suggestion.
Thank you very much for reviewing our manuscript again, Dear Reviewer.
Round 3
Reviewer 2 Report
Ok, many thanks to the authors. I am satisfied by this response completely.